# Explore an open-source value co-creation framework: A multiple case study

Yang Luo[1]*, Yongsheng Jin[1], Yuanmao Ji[2]

1 School of Economics and Management, Beijing University of Posts and Telecommunications, Beijing, China, 2 Ericsson China Academy, Ericsson (China) Communications Company Ltd, Chongqing, China

* loyuang@outlook.com

## Abstract

Open-source communities(OSCs) are gaining significant attention in the current business environment of information technology(IT). More and more IT companies and individuals are exploring how to achieve innovation through open-source collaboration, and value co-creation(VCC) in the OSCs has become a trend. Therefore, it is particularly important to examine the mechanism of OSCs under the background of VCC theory. This study proposes a conceptual framework of open-source value co-creation (OSVCC), which is characterized by openness, sharing, collaboration, and freedom, for understanding the internal mechanisms and contextual conditions in the relationship between OSCs participants. This study constructed a pairwise combined four-category classification model combining the perspectives of the commercialization level (low and high) and the maturity stage (developmental and mature) of the OSCs. Based on the model, this study selects and analyzes four presentive cases of OSCs using a multiple case study approach. Then, this study proposes a framework for OSVCC to identify the crucial factors that promote the successful implementation of innovation and value creation. The OSVCC framework encompasses three primary participants, effective VCC processes, and key open-source principles. This study offers valuable managerial implications for enterprises that plan to participate in OSCs.

## 1 Introduction

The open-source concept, representing the spirit of new capitalism, is a recent economic phenomenon in information technology(IT) that has emerged in the 21st century [1]. The flourishing growth of open-source communities(OSCs) has had a significant impact on the development of the digital economy, such as Linux and Android significantly contribute to the IT infrastructure of the digital economy. Consequently, the value creation of OSCs has attracted the research interest of scholars in the management field. Lin and Maruping [2] considered the sharing of inward knowledge and the reuse of outward knowledge to be a mechanism for creating value in open-source software. Shaikh and Levina [3] suggested that the active participation of software users in healthy OSCs can lead to long-term value creation. Mouakhar and Tellier [4] argued that open-source software is a global phenomenon that

**Data Availability Statement:** All data are publicly available at the following Open Science Framework link: https://osf.io/qy7wr/.

**Competing interests:** The authors have declared that no competing interests exist.

creates value, and it is essential for the world to collaborate in order to protect the values and principles of open-source software.

The value of open-source is undoubtedly realized through the co-creation of community participants. Value co-creation(VCC) theory is a consumer-centered theory in which suppliers and consumers co-create value [5]. It emphasizes that value is co-created through the interaction and cooperation among suppliers, consumers, partners, as well as other stakeholders [6]. However, we find that OSCs are rarely examined under the framework of VCC. There is no supplier-customer relationship in the OSCs, which may not apply VCC theory. OSCs are the unique phenomenon to the IT industry, and the existing VCC framework could not cover such segmented areas. In order to address the above problem of adaptation, Yang and YS [7] introduced the concept of Open-Source Value Co-Creation(OSVCC) which is a VCC system based on common goals and values and following the rules of OSCs, and emphasized that OSVCC is distinct from customer experience [8] and service-dominant(S-D) logic [9]. OSVCC, as a "beyond software" paradigm, prompts the IT companies to shift from product or service focus to engagement focus [10]. Given that OSVCC serves as a critical foundation for the digital economy, it is imperative to develop a conceptual framework for OSVCC.

To bridge the aforementioned theoretical gap, we begin with a pairwise combined four-category classification model along the dimensions of maturity and commercialization. Building on this model, we select four typical cases and detail our case analysis, and then propose a conceptual framework of OSVCC. We discover the relationship between the participants of OSVCC is no longer a supplier-customer relationship, but a triangle relationship, and the decision of either party may affect the stability of the triangle relationship. This framework stresses the importance of open-source collaboration and organization learning in OSCs. Our findings bridge the theoretical gap, provide valuable insights for understanding the VCC theory in OSCs, and guide enterprises and individuals involved in OSCs. Finally, we discuss both theoretical and managerial implications.

## 2 Theoretical underpinnings

In this section, we review the relevant theories that will be used in this paper. Firstly, we compare the differences between OSVCC and other VCC research perspectives and review the past research of VCC framework. Then, we introduce the business ecosystem theory and open innovation theory related to OSCs which will be the theoretical basis for the construction of the classification model in the next section.

### 2.1 Value co-creation in open-source community

After more around 20 years of development, VCC theory has formed two main research perspectives [11]. One research perspective of the VCC theory is based on customer experience, and it mainly focuses on the creation of personalized customer experience value through the interaction between customers and suppliers [8, 12]. However, there are only participants and contributors, and no clear customer-supplier distinction in the OSCs. Another one is based on S-D logic [9, 13], including service logic, service science, and service ecosystem [14]. The S-D logic emphasizes that service exchange is the core of value creation, and value is determined and co-created by customers [15, 16]. The S-D logic is an economic model proposed based on the macro perspective of economic development and evolution mode [17] and rarely guides individuals and enterprises from the micro perspective of enterprise strategic management [18].

Therefore, it is difficult for the two mainstream research perspectives of VCC to be directly applied to the research of OSCs, but a few scholars have made some attempts. Battistella, *et al.* [19] emphasized that VCC in OSCs relies heavily on the contributions of user innovators, and

**Table 1. Comparison of the VCC research perspective.**

|  | Customer experience | S-D logic | OSVCC |
|---|---|---|---|
| **Perspective** | Enterprises' strategy and competition | Economic development and evolution | Open-source community |
| **Scope** | Intermediate | Marco | Micro |
| **Value-based** | Personalized experience | Service-use | Collaboration |
| **Co-creators** | Customers and suppliers | Customers and suppliers | Open-source participants |
| **Initiative** | Customers | Suppliers | Everyone |

Note: The authors compiled this table.

the sustainability of enterprise participants depends on the reputation of the firm and the improvement of working knowledge. Nagle [20] elaborates on the idea of gaining a competitive advantage by participating in the VCC of open-source software. It will not only bring multiple benefits and competitive advantages to individuals or organizations but also can promote social responsibility and citizenship, bringing more social value to an individual or organization [20, 21]. Based on reviewing nearly two decades of articles on open-source software and value creation, Yang and YS [7] proposed OSVCC which is a VCC ecosystem characterized by openness, sharing, collaboration, and freedom, with open-source code as the carrier and multiple participants collaborative innovation. In OSVCC, open-source participants have the opportunity to actively engage in innovation, designing, programming, as well as decision-making regarding software development [7]. Compared to customer experience and S-D logic, OSVCC is more specific, focuses on VCC in OSCs, and emphasizes contributors co-create value through collaboration in OSCs(see Table 1).

Scholars have shown sustained interest in researching VCC models for the past 20 years, resulting in some research findings. Prahalad and Ramaswamy [12] proposed the DART model, which categorized the structure of VCC into four subdimensions: dialogue, access, risk, and transparency. After conducting in-depth interviews with executives from 18 large organizations, Payne, *et al*. [15] proposed a VCC process framework that divided the process into three parts: customer VCC, supplier VCC, and encounters. Since then, scholars have sequentially refined the framework for the process of VCC in knowledge-intensive industries [22] and the mechanism for ecosystems of innovation [23]. However, the existing VCC framework does not target the OSCs. So, it is not suitable for those enterprises and individuals who wish to participate in the OSCs. Therefore, it is essential to construct an OSVCC framework, which has significant theoretical and practical implications for understanding the mechanism of value creation within OSCs. The framework could guide programmers who want to participate in OSCs and aid IT enterprises that plan to join OSCs in formulating open-source strategies.

## 2.2 Business ecosystem

The concept of a business ecosystem extends the idea of "ecology" in biology to the field of business. It highlights the symbiotic coexistence of enterprises and stakeholders, as well as the ability of enterprises to lead and cultivate the ecosystem. Moore [24] initially proposed the concept of the business ecosystem and defined the business ecosystem as an "economic union based on organizational interaction". Later, in order to further clarify the internal structural characteristics and evolution mechanism of the business ecosystem, Moore [25]pointed out that the business ecosystem is a dynamic structural system composed of organizations or groups with certain interests. After that, many scholars have devoted themselves to the study of business ecosystems. Iansiti and Levien [26] proposed the concept of niche to illustrate the

structural characteristics of the business ecosystem. From the perspective of enterprise network, Den Hartigh, *et al.* [27] argued that a business ecosystem is a network of interdependent suppliers and customers around a certain core technology. Senyo, *et al.* [28]introduced that the digital business ecosystem is an extension of the business ecosystem, and the core enterprises could create value by building the digital business ecosystem. OSCs are a typical digital business ecosystem, which are online communities around core technology and follow the law of the business ecosystem. Therefore, it is appropriate to analyze OSCs from the perspective of business ecosystem theory. Business ecosystem theory provides a dynamic perspective that helps enterprises position themselves in an open-source environment, as well as establish collaborative relationships with other members of the ecosystem.

### 2.3 Open innovation

For a long time, enterprises have firmly believed that internal research and development capabilities are valuable strategic assets, emphasizing that successful innovation requires strong control. However, Chesbrough [29] held a different view and proposed the theory of open innovation through research on high-tech companies, advocating for companies to intentionally harness both internal and external innovation. Baldwin and Von Hippel [30] explored the paradigm shift in innovation patterns by comparing three innovation paradigms: single-user or company innovation, open collaborative innovation, and producer innovation, emphasizing the importance of new business models that support user innovation and open collaborative innovation. Subsequently, Chesbrough and Bogers [31] expanded the concept of open innovation and proposed that open innovation can be realized through different types of profit and non-profit mechanisms, which is a distributed innovation process that crosses organizational boundaries and effectively manages knowledge flows. OSCs are distributed innovation paradigms formed through open-source collaboration, and OSCs' participants achieve innovation through knowledge flow across organizational boundaries. Therefore, open-source communities can be analyzed in the context of open innovation theory. Open innovation theory encourages IT enterprises to accelerate innovation processes through the sharing of code, knowledge, and best practices. By participating in OSCs, enterprises could absorb external innovation resources, and at the same time enhance their influence and competitiveness in the commercial market by contributing their own innovation achievements.

## 3 Research design

This section describes the combined use of the business ecosystem theory and the open innovation theory to develop a pairwise combined four-category classification model of OSCs. Subsequently, we outline the research method and selection of appropriate cases, according to the classification model.

### 3.1 Classification model

According to the research questions, this paper mainly draws on the business ecosystem theory and open innovation theory to build up a classification model of OSCs. This classification model is to clarify the situational differences and conditions of OSCs, divides all OSCs into four categories, and provides a theoretical basis for case analysis.

Moore [24] introduced the concept of the business ecosystem and categorized its development into four stages: birth, expansion, leadership, and self-renewal. OSCs are ecosystems in which multiple participants collaborate to create and share value through innovative collaboration. This ecosystem also experiences a gradual process of development and maturity. Based on the research of open source software development process, Petrinja, *et al.* [32] proposed the

open-source maturity model (OMM) and divided it into three maturity levels: basic, interme-diate, and advanced. Kuwata, *et al*. [33] also categorized the development of OSCs into five stages: initial, managed, defined, quantitatively managed, and optimized. As technology advances and market needs change, the participants in OSCs continuously adapt their strate-gies to the changing ecosystem environment. In the early stages, the participants could focus more on the sharing of technology and the rapid iteration of code [34]. As ecosystems mature, they could shift to a greater focus on collaboration and standardization to ensure compatibility and interoperability between different components. With the increasing influence, partici-pants could focus more on business model innovation and explore how to make money by providing value-added services, customizing solutions, or building specific platforms [19]. At the same time, they will pay close attention to user feedback and market trends, so as to timely adjust products and services to meet user needs. Ultimately, as the ecosystem stabilizes, partici-pants are likely to place more emphasis on maintaining and optimizing the existing system, while continuing to explore new technological areas to keep the ecosystem dynamic and com-petitive [2]. This study introduces a two-stage model for OSCs: the developmental stage and the mature stage. The developmental stage aligns with Moore [24] proposed concept of birth and expansion, while the mature stage requires stability and the production of reliable open-source software products.

West and Lakhani [35] were the first to introduce the idea that the research paradigm of open innovation applies not only to enterprises but also to the study of OSCs. Morgan and Fin-negan [36] argued that open-source software represents the most developed form of open innovation, and highlighted the benefits of engaging in OSCs, such as reducing costs through inward participation and creating value through outward participation. Chesbrough [37] believes that to obtain innovation profits, enterprises not only need technological innovation but also need appropriate business models to commercialize the technology. Enterprises can generate value by commercializing technology, and make strategic choices regarding open-source technologies [38]. Chesbrough and Schwartz [39]highlights the importance of co-devel-opment partnerships in business models, which are mutually beneficial working relationships between two or more parties aimed at creating and providing new products, technologies, or services. This partnership could allow OSCs to be commercialized without waiting for the technology to mature. The enterprise that dominates the community influences the commer-cialization of the community, leading to varying degrees of commercialization. When the dominant enterprise chooses to start commercialization, it usually needs to consider its own business situation[40] and the co-development relationships between participants in the com-munity[41]. Therefore, we classify OSCs into two grades in the commercialization dimension: low and high.

According to the above analysis, OSCs can be classified into two dimensions: maturity and commercialization. In the developmental phase, OSCs have constructed a collaborative ecosys-tem and continue to attract a variety of participants. In the mature phase, OSCs have estab-lished stable cooperation among participants, gained influence in specific areas, and even become leaders in segment technology areas. The degree of commercialization is usually deter-mined by the founder of the OSC. Founders who choose low commercialization often have alternative sources of revenue and strong financial resources that prioritize the long-term ben-efits of their open-source strategy. On the other hand, founders who choose to be highly com-mercial often focus solely on open-source projects or are enticed by significant market profits from open-source commercialization. We divide all OSCs into four categories through the two dimensions of maturity and commercialization. The resulting classification model for the OSCs is presented in Fig 1.

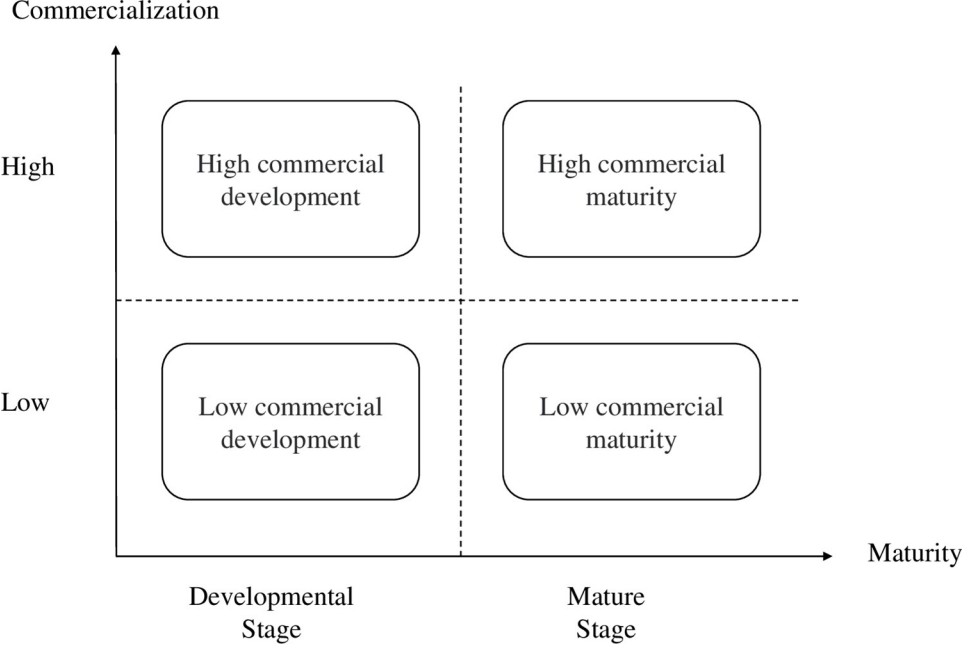

**Fig 1. The classification model of OSCs.**

## 3.2 Research method

This study aims to investigate a framework for OSVCC to identify the crucial factors in OSCs that promote the successful implementation of innovation and value creation. The case study method is an effective approach for constructing and validating theories [42] and does not rely on previous empirical evidence or existing literature. It is suitable for addressing the "how" problems and exploring new phenomena that have emerged in practice. Therefore, the research objective aligned well with the case study method. In contrast to the use of a single case study, the multiple case study method allows for a more comprehensive understanding of the mechanism of how a phenomenon occurs relative to the cases studied. Additionally, it aids in expanding the existing theoretical system and constructing a new theoretical framework [43].

The conclusions drawn from a multiple case study are more general and persuasive, facilitating the extraction of stable and universal propositions from OSCs. Moreover, the multiple case study method effectively reveals the relationships between elements and also provides a deep understanding of the mechanisms of OSCs. Therefore, this study employs multiple case study methods to enhance the generality and validity of the conclusions and identify the crucial factors of OSVCC.

## 3.3 Case selection

This study adopted the theoretical sampling method for case selection, which involved selecting and collecting the raw data according to the research objectives. This study primarily investigated the operational mechanism through the dimensions of maturity and commercialization of OSCs to develop a conceptual framework. Four OSCs were selected on the basis of the research questions and analytical method, using the following screening criteria:

1. The selected cases must be OSCs that have been continuously operating since their establishment and have organized offline community activities at least once every six months;

**Table 2. The background information for the selected OSCs.**

|  | TiDB | OpenHarmony | OpenAnolis | PaddlePaddle |
|---|---|---|---|---|
| **Setup** | 2015 | 2020 | 2021 | 2016 |
| **Founder** | PingCAP | Huawei | Alibaba | Baidu |
| **Domain** | Database | Mobile OS | Server OS | AI |
| **Headquarters** | Beijing | Shenzhen | Hangzhou | Beijing |
| **Employees** | <300 | ≈200,000 | ≈240,000 | ≈40,000 |
| **Foundation** | CNCF | OpenAtom | None | None |

Note: The authors compiled this table using publicly available information.

2. The selected cases must be typical and representative OSCs from different domains with global influence, organizing or participating in at least one global technology workshop or summit in this domain every year;

3. The selected cases must have been established for more than two years and have sufficient online information for research.

Based on the above criteria, the four OSCs selected were TiDB, OpenHarmony, OpenAnolis, and PaddlePaddle. Table 2 presents the background information for each project.

### 3.4 Data collection

The data were collected from multiple sources, ensuring the reliability and validity of each case study through mutual verification [43]. This paper primarily collected data from online sources and interviews. The online data collection process occurred between September 2021 and September 2023 and comprised internal and external data. The internal data encompasses content from the community's official website, official WeChat account, GitHub, Gitee, and corporate annual reports. The external data includes third-party research reports, news, and literature databases. The authors thoroughly reviewed and compared these data, and continuously replenished during the analysis. Additionally, cross-verification was conducted using multiple data sources to ensure the reliability, authenticity, and validity of the research. The data sources for each case study are displayed in Table 3.

The interviews were conducted via video conference during March 2024, and the interview data were used for the theoretical saturation. Using a semi-structured protocol, the interviews were conducted with a total of 14 participants and aimed at collecting the opinions of the respondents on the draft framework. The interviewees came from different organizations with different roles in the OSCs, and each interview involved at least two authors. Each interview lasted between 30 and 40 minutes, and the iFLYREC software was used to convert speech to text. After the interviews, the two authors cross-checked the audio recordings and texts to

**Table 3. Online data sources.**

|  | TiDB | OpenHarmony | OpenAnolis | PaddlePaddle |
|---|---|---|---|---|
| Internal materials | Official website, official WeChat account, and GitHub | Official website, official WeChat account, Gitee, and annual report of Huawei | Official website, official WeChat account, and Gitee | Official website, official WeChat account, GitHub, and annual reports of Baidu |
| External materials | Third-party reports, news, and literature databases | Third-party reports, news, and literature databases | Third-party reports, news, and literature databases | Third-party reports, news, and literature databases |

Note: The authors compiled this table using publicly available data sources.

**Table 4. Interview information.**

| OSCs | Interviewee | Organization | Position | Duration |
|---|---|---|---|---|
| TiDB | S1 | PingCAP | Marketing Specialist | 30 minutes |
| | S2 | Dmall | Database Administrator | 30 minutes |
| | S3 | N/A | Individual Developer | 40 minutes |
| OpenHarmony | S4 | OpenAtom | Operation Consultant | 30 minutes |
| | S5 | Huawei | Open-Source Specialist | 30 minutes |
| | S6 | Hoperun | Software Engineer | 40 minutes |
| | S7 | N/A | Individual Developer | 40 minutes |
| OpenAnolis | S8 | Alibaba | Open-Source Specialist | 30 minutes |
| | S9 | Inspur | Software Engineer | 40 minutes |
| | S10 | Kylin | Software Engineer | 40 minutes |
| PaddlePaddle | S11 | Baidu | AI Ecosystem Manager | 30 minutes |
| | S12 | CMCC | AI Algorithm Engineer | 30 minutes |
| | S13 | N/A | Individual Developer | 40 minutes |
| | S14 | NA | Individual Developer | 30 minutes |

Note: All interviewees requested anonymity.

ensure the accuracy and credibility of the information. The interview information is displayed in Table 4.

## 3.5 Data analysis

This paper recruited multi-level coding to analyze the online data, including internal and external data of the case. The coding process was independently completed by one author, and the other author was responsible for review. Then, the two authors discussed and finalized the coding. The data analysis process adopted the grounded theory method, which decomposes and conceptualizes the data, and then forms a certain logical relationship diagram[44]. It was mainly divided into the following three steps [45]: In the first step, open coding was used to classify the data by source in order to identify categories and sub-categories. The second step is axial coding, which is the process of conceptualization of the core elements. The third step is selective coding to conceptualize the core elements of the OSCs.

In order to determine the type of OSC belongs, this study systematically screened a vast number of primary materials, and identified the relevant information, using a four-step process as follows:

1. Examining the third-party research reports in chronological order, extracting the content of the relevant open-source cases, and analyzing the ecological maturity of the cases;

2. Determining the ecological maturity of the cases by analyzing the objective indicators such as stars, pull requests (PR), and forks on GitHub and Gitee, in conjunction with the results of the first step;

3. Determining the degree of commercialization of each case by analyzing information about its business-related cooperation from the official website, official WeChat account, corporate annual reports, and news;

4. The form and content of VCC in each case were combined and summarized, and the cases were compared repeatedly. Discussion, supplementation, and optimization achieved a consensus determining the ecological maturity and degree of commercialization of each case.

In order to minimize bias resulting from subjective factors, it is essential to utilize objective data for clear and unambiguous measurements of the key concepts during the process of analyzing the cases. To assess the OSCs' maturity, this study focuses on the number of stars, PR, and forks acquired by each case on the GitHub and Gitee platforms. To confirm the level of commercialization, this study examines the number of users, commercial cases, and business partnerships, and focuses on considering the business cases implemented in critical business systems. These measures ultimately affect the determination of the maturity stage and the commercialization of the cases.

## 4 Case analysis

This section provides a comprehensive analysis of each case, based on the four categories of OSCs in the classification model: high commercial maturity(HCM), high commercial development(HCD), low commercial development (LCD), and low commercial maturity (LCM). Following the case study analyses, the results show that three types of participants can be identified as the primary participants in the OSVCC.

### 4.1 High Commercial Maturity(HCM)

TiDB, an open-source relational database, was independently designed and developed by PingCAP in 2015. It was announced as being open-source on the day of its release and has been adopted by over 2,000 companies in various industries as a production environment. Currently, the TiDB project has garnered over 34,000 stars on GitHub and has attracted more than 2,100 open-source contributors. It has established itself as a top-tier database project in the global IT field. TiKV, the storage engine of TiDB, graduated from the Cloud Native Computing Foundation (CNCF) in September 2020. It was the second Chinese original open-source project to graduate from CNCF.

After 8 years of rapid development, the TiDB community has become highly active, with frequent updates of both the English community on GitHub and the Chinese community on its official website. Currently, it has achieved an impressive milestone with 34,000 stars and 96,000 PR, which is remarkable for a database OSC. Therefore, the TiDB can be considered as a mature stage of OSC. In terms of commercialization, TiDB has also experienced rapid growth, attracting over 2,000 corporate users from important industries in sectors such as finance, the internet, and government. These users include well-known financial institutions, such as Ping An Insurance (the largest insurance company in China) and the Bank of China (one of the largest banks in China), renowned internet companies such as Meituan (one of the most popular APPs in China) and Zhihu (the largest Chinese knowledge-sharing community in the world), as well as significant state-owned corporations, such as China Telecom and State Grid. Consequently, this study identifies the TiDB community as a highly commercial and mature OSC.

The TiDB community consists of a wide variety of participants, who are mainly divided into developers and users. PingCAP, as the primary developer and contributor to TiDB, has assumed the role of the community leader. It has achieved this leadership position by actively contributing code, sponsoring various community activities, and influencing the technical direction. The users of TiDB primarily consist of enterprise users, encompassing both free and paid users. Paid users can be further categorized into subscription-based users and on-demand cloud users, depending on their terms of payment. The TiDB community encompasses both enterprise developers and individual developers, although the number of enterprise developers is relatively small. Nevertheless, individual developers play crucial roles within the community, with some having made continual contributions for nearly 8 years. To facilitate the users'

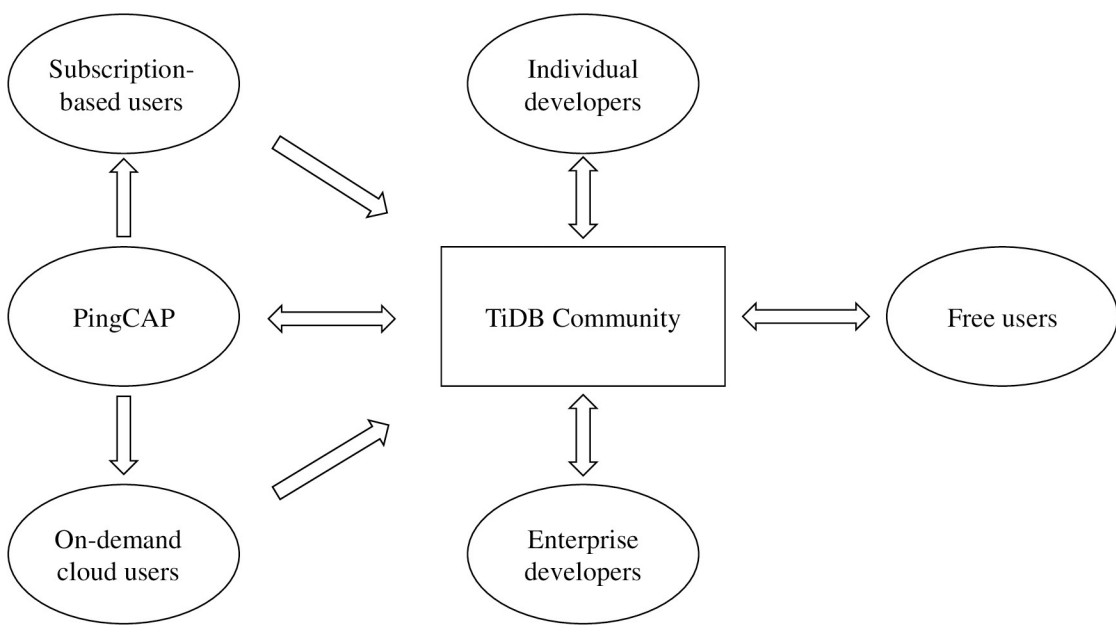

**Fig 2. Relationships between the participants of the TiDB community.**

engagement, the TiDB community has established a user organization called the TiDB User Group (TUG), which actively encourages users to provide feedback. Relationships between the participants of the TiDB community. The relationship between the participants of the TiDB community is shown in Fig 2.

## 4.2 High Commercial Development(HCD)

OpenHarmony is an open-source operating system that was donated by Huawei to the OpenAtom Foundation in 2020. It primarily targets consumer and industry markets for terminal devices and aims to provide a unified operating system for various smart devices. Right now, the Harmony OS, which is based on OpenHarmony, has become the third-largest operating system for smart devices in the world. OpenHarmony holds the top position in the Gitee Activity Index(GAI) and has 51 co-building organizations and over 5000 contributors. It is highly sought after as one of the most popular OSC in China.

The OpenHarmony community was established in September 2020 and has experienced rapid growth. The number of contributors increased from 1,000 in 2021 to 5,000 in 2022. Thus far, it has achieved excellent results on Gitee, with 22,000 stars and 55,000 forks. However, there is still a significant gap compared with mature smart operation system communities, such as Android, because only a handful of handset manufacturers have developed operating systems based on OpenHarmony. Therefore, this study considers that the OpenHarmony project is still in the developmental stages of OSC. Despite being established only 2 years ago, the OpenHarmony community has already achieved significant milestones in terms of commercialization. On the one hand, Huawei, as a significant player in the global consumer electronics market, saw a surge of 180 million new terminal devices using Harmony OS in 2022. These users have become the cornerstone of the OpenHarmony ecosystem. On the other hand, enterprise developers have issued multiple commercial operating systems derived from the Open-Harmony system, which have been implemented across a wide range of smart devices and

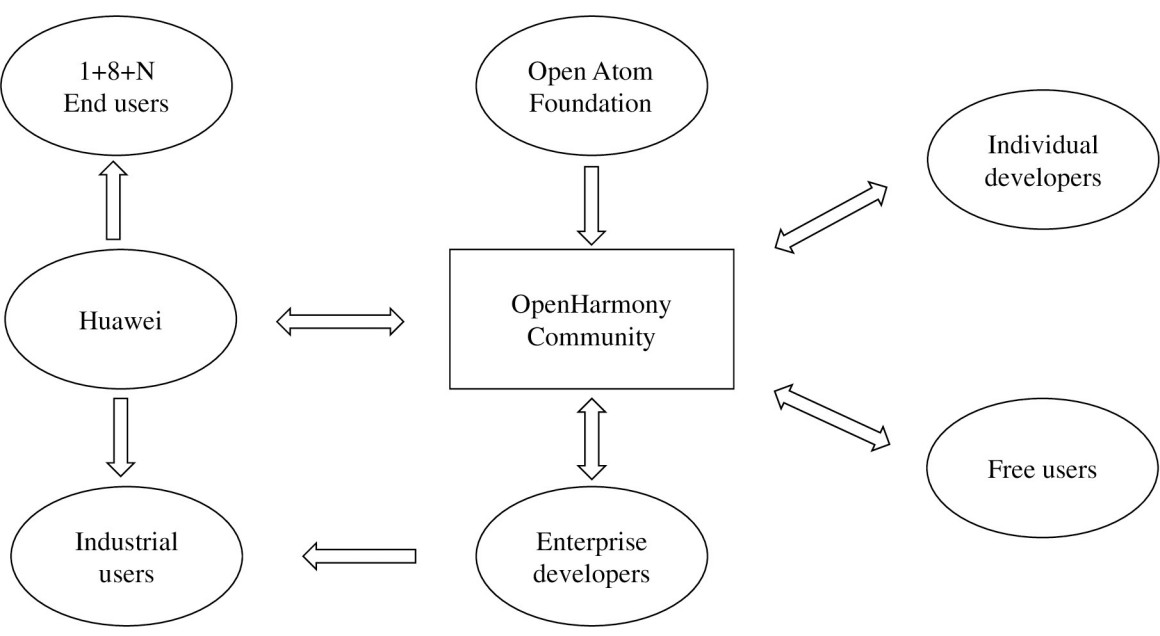

**Fig 3. Relationships between the participants of the OpenHarmony community.**

industry solutions. Therefore, this study identifies that the OpenHarmony community is a highly commercial and developmental OSC.

The OpenHarmony community comprises the OpenAtom Foundation, the developers, and the users. The OpenAtom Foundation, as a nonprofit public welfare organization, has assumed the responsibility for the daily management and organization of the community's activities. Huawei, as one of the founders of the OpenAtom Foundation and the primary contributor to the OpenHarmony project, plays a prominent role as a corporate developer within the community. Huawei actively contributes code, sponsors community activities, and engages in the governance of the community, thus taking a leading position in the OpenHarmony community. Apart from Huawei, the community of enterprise developers comprises 51 companies, including Kaihong, Hoperun, Pateo. Collectively, these enterprise developers have launched more than 100 commercial products, including software and hardware, built on the OpenHarmony platform. Additionally, some individual developers and free users are actively participating in the community's co-construction. The relationship between the participants of the OpenHarmony community is shown in Fig 3.

## 4.3 Low Commercial Development(LCD)

The OpenAnolis community is an upstream Linux distribution community that was initiated by Alibaba in October 2020. Its main objective is to develop an open-source operating system called Anolis OS. Anolis OS supports various computing architectures, optimizes for cloud scenarios, and is compatible with the software ecosystem of CentOS.

The goal of Anolis OS is to provide developers and operations personnel with a stable, high-performance, secure, reliable, and open-source operating system, whose main users are the original CentOS users. Currently, OpenAnolis has garnered only 300 stars on Gitee and is in the donation period of the OpenAtom Foundation. Thus, this study posits that OpenAnolis is in the developmental stage of OSC. In terms of commercialization, Anolis OS was already widely used within the Alibaba Group before it became open-source software. The

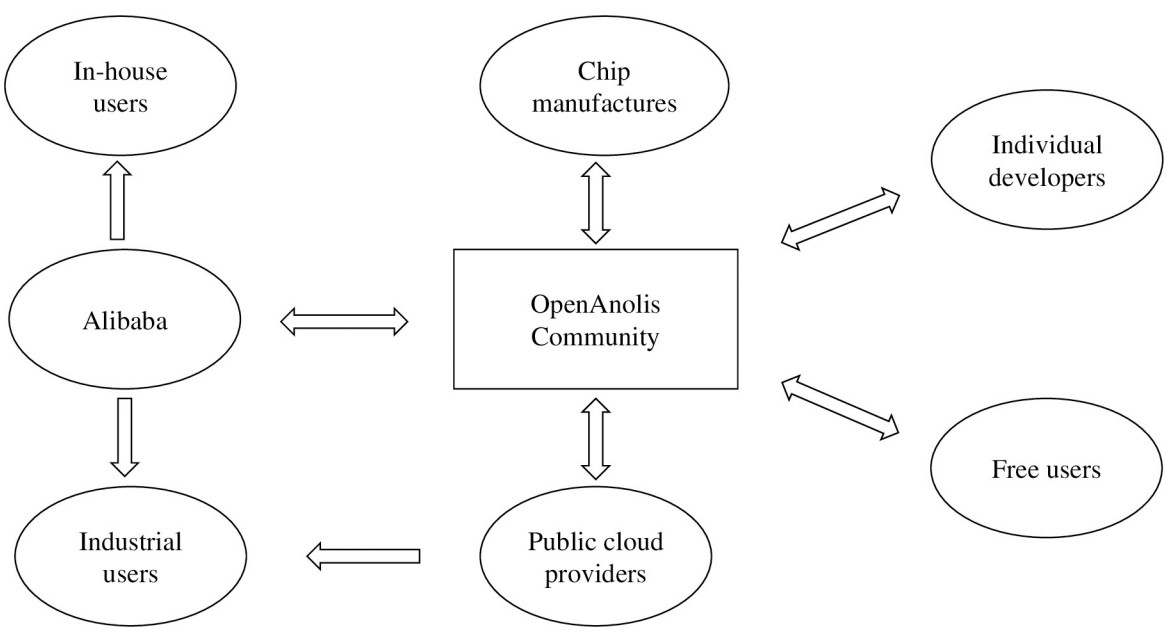

**Fig 4. Relationships between the participants of the OpenAnolis community.**

OpenAnolis 2022 whitepaper revealed that the majority of commercial cases are derived from the in-house applications of Alibaba Group, with fewer instances of external commercial utilization. Consequently, this study classifies the OpenAnolis community as a low commercial and developmental OSC.

The OpenAnolis community was established 2 years ago and is pursuing a distinct pathway compared with other Linux communities. In contrast to the conventional dual-helix model followed by operating systems providers and chip manufacturers, the OpenAnolis community actively incorporates public cloud service providers, such as Alibaba Cloud, China Telecom Cloud, and China Mobile Cloud, forming a collaborative model with chip manufacturers, referred to as the "Iron Triangle". Alibaba, as the primary contributor and user of Anolis OS, has assumed a leading role. Enterprise developers, including chip manufacturers, public cloud providers, and system integrators, cooperate to contribute to the community. The chip manufacturers primarily focus on adapting Anolis OS, while the public cloud providers serve as promotional channels. Additionally, a small number of individual developers participate in the collaborative development of the OpenAnolis community. The relationship between the participants of the OpenAnolis community is shown in Fig 4.

## 4.4 Low Commercial Maturity(LCM)

PaddlePaddle, launched by Baidu in 2015, was the first open-source deep-learning platform in China. PaddlePaddle integrates core frameworks, basic model libraries, end-to-end development kits, and a wide range of tool components. It enables developers to efficiently implement artificial intelligence (AI) ideas, develop new AI applications, and support various industries in achieving industrial intelligence upgrades.

The PaddlePaddle community has been established for 8 years and currently has 3600 followers on GitHub, with over 80,000 stars. PaddlePaddle is the most active Chinese OSC on GitHub and is one of the most widely used AI frameworks in China. Therefore, this study asserts that PaddlePaddle is a mature OSC. However, in terms of commercialization, the

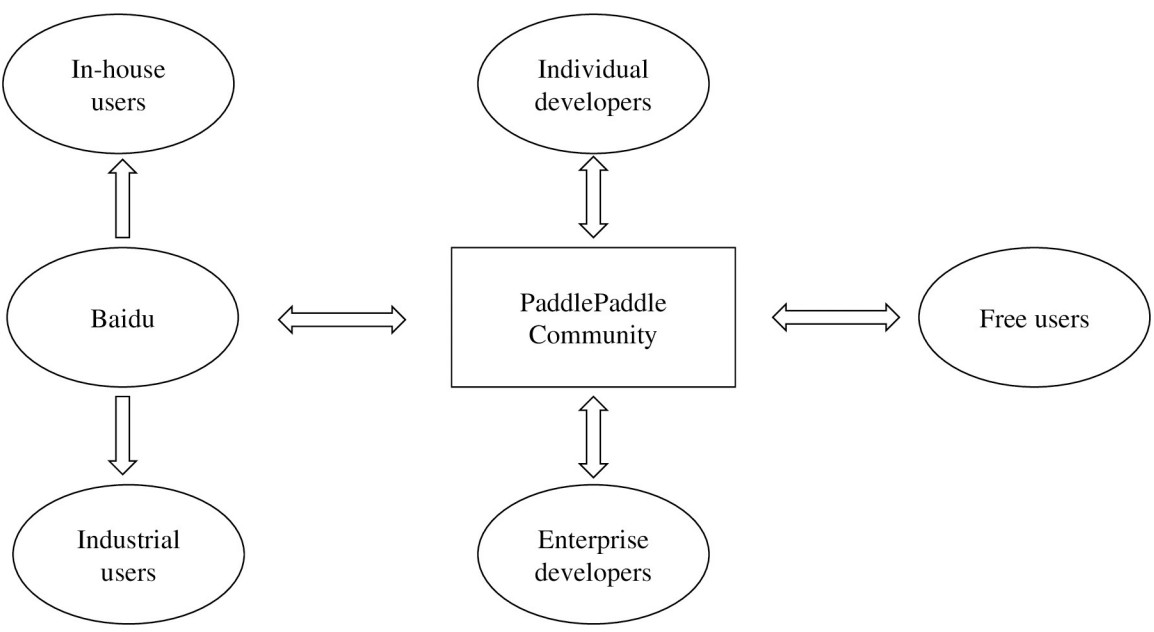

**Fig 5. Relationships between the participants of the PaddlePaddle community.**

majority of PaddlePaddle's users are utilizing the free community version, and there is a limited number of paid users. Despite Baidu's efforts to expand its commercial development suite, the large-scale commercial AI system in China continues to be predominantly deployed on other open-source frameworks, such as TensorFlow and PyTorch. Consequently, this study classifies the PaddlePaddle community as a low commercial and mature OSC.

The PaddlePaddle community comprises many developers and users. Baidu, as the initiator and primary contributor of the community, has assumed a leading role. Additionally, Baidu is the largest commercial user of PaddlePaddle. PaddlePaddle community encompasses a significant number of individual developers, primarily consisting of enthusiasts, university students, and researchers. These individual developers are also the users of the software. While contributing to the community, individual enthusiasts also have access to the latest version of the software at no cost. Moreover, a small number of enterprise developers are in the community, representing various industries, who utilize PaddlePaddle to enhance their respective companies' productivity. The relationship between the participants of the PaddlePaddle community is shown in Fig 5.

## 4.5 Comparative analysis of primary participants

The realization of VCC varies depending on the differences in the participant's relationships. Hence, it is essential to consider the variations between the participants. According to the analysis of the aforementioned four cases, this study posits that the following three roles are the primary participants in OSCs:

1. The dominant participant(DP), which is usually the initiator, exerts influence over the development of the OSCs. The DP plays a pivotal role in driving OSCs, reaping significant social and commercial benefits [46];

2. Enterprise participants (EPs) consist of developers and users from enterprises in the OSCs. Enterprise developers (EDs) primarily consist of upstream and downstream companies in

the supply chain of the DP. Enterprise users (EUs) typically prioritize the performance and usability of open-source products [46], and EUs with development capabilities are also EDs;

3. Individual participants (IPs) consist of independent developers and users who are not affiliated with corporations. Generally, individual developers(IDs) also serve as individual users (IUs), whose engagement in OSCs is motivated by personal interests or career development [46–48].

In the TiDB case, the DP, PingCAP, has profited from a high degree of commercialization, so it continues to provide sufficient support for the community and actively participates in OSVCC. Mature communities could provide rapid response and powerful supporting facilities for EUs, so EUs actively participate in OSVCC; however, higher maturity suppresses the space in which enterprises can participate and inhibits their willingness to develop, so the number of EDs in the community is small. IUs could get more effective support in mature communities, so they are willing to actively participate in contributions; at the same time, the participation of IDs in well-known open-source projects brings them prestige and job opportunities, so IPs actively participate in OSVCC.

In the OpenHarmony case, although the DP, Huawei, has benefited from commercialization, the community is still in the developmental stage and needs to attract more participants to make positive contributions. The community is not mature yet, but it has certain commercialization prospects, which means market opportunities, so EDs and EUs actively participate in OSVCC. Higher commercialization prospects could attract IDs, but the community is still in the developmental stage and the supporting facilities are not ready, resulting in fewer IDs and IUs joining in OSVCC.

In the case of OpenAnolis, the open-source software has already been used internally for several years, and the DP, Alibaba, could contribute accumulated experience to the community to accelerate OSVCC collaboration. Since the community is still in its infancy and the commercial prospects are unclear, EUs are usually not actively involved in OSVCC. However, considering the huge internal use demand of DP, EDs join in OSVCC collaboration under the influence of DP. In this case, IDs and IUs are mainly participating in the community out of interest, and a small quantity.

In the case of PaddlePaddle, the software has widely and fully utilized the DP's internal and external resources, but there are shortcomings in commercial promotion. EDs prefer to use the mainstream AI framework in large-scale commercial use scenarios and try to participate in OSVCC in small-scale scenarios. Due to personal interest, the popularity of the community, and the complete supporting facilities, IPs and IUs actively participate in OSVCC collaboration. The comparison of participants in the four cases is shown in Table 5.

**Table 5. Participants in OSCs.**

| Participants | | HCM (TiDB) | HCD (OpenHarmony) | LCD (OpenAnolis) | LCM (PaddlePaddle) |
|---|---|---|---|---|---|
| DP | | PingCAP | Huawei | Alibaba | Baidu |
| EPs | EDs | Few | Hoperun, Pateo | Intel, Uniontech | Few |
| | EUs | Meituan, Zhihu | Huawei, Midea | Alibaba | Baidu, JD |
| IPs | IDs | Numerous | Few | Few | Numerous |
| | IUs | Numerous | Few | Few | Numerous |

Note: The authors compiled the table using the information from the official website of the OSCs.

**Table 6. The process of OSVCC.**

| No. | Process | Participants |
|---|---|---|
| **1** | DP's OSVCC process | DP |
| **2** | EPs' OSVCC process | EPs |
| **3** | IPs' OSVCC process | IPs |
| **4** | DP and EPs encounter process | DP and EPs |
| **5** | EPs and IPs encounter process | EPs and IPs |
| **6** | IPs and DP encounter process | IPs and DP |

## 5 Findings

This section describes the main components of the new conceptual framework based on the principles of VCC and reveals the OSVCC mechanism. This new framework is an extension of the classic VCC framework.

### 5.1 Conceptual framework of OSVCC

As previously described, this study identifies that all four selected cases consist of three key types of participants: DP, EPs, and IPs. This deviates from the typical scenario of VCC that involves only suppliers and customers. Consequently, the framework of OSVCC needs to consider the three participants and their interrelationships, resulting in a more intricate architecture. The classic VCC framework proposed by Payne, *et al.* [15] comprises three parts: the customers' VCC process, the suppliers' VCC process, and the encounter process. Therefore, a framework comprising six processes, based on the participants and their relationships, is presented in Table 6.

This paper provides an overview of the framework structure and explains the components in detail, as shown in Fig 6. In this framework, the participants in the OSVCC are more specific

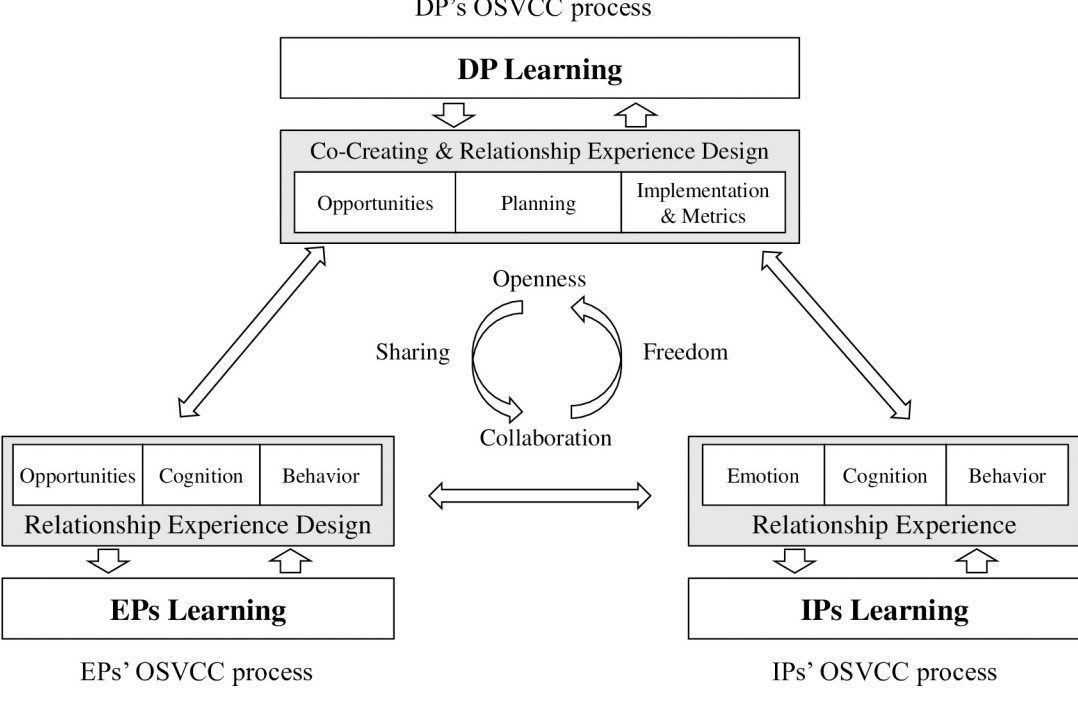

**Fig 6. Conceptual framework of OSVCC.**

and explicit compared with the classical VCC framework. The DP typically initiates open-source projects, but the DP neither owns the final product nor has an employer-employee relationship with the other participants [49]. DP serves as the primary driving force behind the OSCs' development and reaps the most significant social and commercial benefits. EPs have a complementary role in the open-source community, and their relationship with the DP is delicate [41]. EPs primarily comprise the upstream and downstream enterprises of the DP's value chain. IPs frequently serve as the most active participants in the framework, deriving great satisfaction from contributions [50]. IPs join the community on the basis of their interests or career development. These three primary participants and the interactions between them constitute the crucial factors of OSVCC. Participants collaborate and work interdependently, while maintaining their independence to create value, guided by the principles of openness, sharing, collaboration, and freedom.

Theoretical saturation means that the collected data will no longer generate new categories or theoretical discoveries [51]. This paper used a semi-structured interview and 14 participants were interviewed for their opinions on the framework of OSVCC. The two authors continued to analyze the interview data in a multilevel coding method. The results of the analysis showed that the concepts and categories obtained have been included in the existing categories, and no new concepts and categories have been generated. Therefore, this paper believes that the conceptual framework of OSVCC has a good theoretical saturation and can reflect the nature of the phenomenon.

## 5.2 The OSVCC process

**5.2.1 The DP's OSVCC process.** The DP is usually an IT company with specific strengths, and its OSVCC process primarily involves breaking down organizational boundaries and engaging extensively with the OSCs. Payne, *et al.* [15] believe that supplier achieves VCC by designing and delivering relationship experiences and promoting organizational learning. Hence, in the context of OSVCC, the VCC process of the DP encompasses four stages:

1. Usually, the DP is at the forefront of the IT industry, is adept at capturing technological and marketing changes, and is the first to identify potential opportunities and propose possibilities to create value;

2. After conducting a thorough analysis, the DP decides to adopt an open-source strategy that considers both the internal and external company environments. The DP develops a VCC plan that revolves around the open-source strategy;

3. The DP implements the VCC plan and actively communicates with other participants, establishing a consensual value proposition and collaborating on innovative activities;

4. The DP evaluates the results of the implementation, conducts a joint analysis of the efficiency and effectiveness of VCC, and strives for continuous improvement.

The DP's OSVCC process involves formulating open-source strategies, establishing open organizational structures, and attracting both EPs and IPs to join the OSVCC in order to collectively build an ecosystem. The DP also facilitates organizational learning through interactions with the other participants. Collaboration with other participants will bring new knowledge, resources, and opportunities for innovation to the DP, and helps the DP have a better understanding of the market's demands and co-creating value. The DP's learning enables the DP to continuously enhance its capabilities and adaptability, leading to improved VCC and business performance.

**5.2.2 The EPs' OSVCC process.** The EPs typically assign employees to engage in OSCs, contributing to the development, maintenance, and promotion of open-source projects, while also providing feedback to the OSCs regarding the use of open-source software. The EPs play dual roles as both supplier and customer in the OSVCC framework, compared with the classic framework proposed by Payne, *et al.* [15]. The EPs actively engage in designing and delivering relationship experiences, continuously contribute to the OSCs through their cognitive and behavioral efforts, and facilitate organizational learning.

1. EPs are generally companies in the DP's value chain, either upstream or downstream. This positioning enables EPs to effectively identify and seize opportunities for VCC that are initially discovered by the DP;

2. EPs confirm the existence of opportunities through multiple communications with the DP, utilizing their cognitive abilities. EPs then formulate enterprise open-source strategies and engage in OSVCC;

3. EPs exhibit distinct behavior compared with the other two types of participants. EPs have a dual motivation: on the one hand, EPs aspire to actively participate in OSVCC and contribute to the community; on the other hand, EPs aim to maximize their own interests, which may lead to "free riding."

The organizational learning of EPs is similar to that of the DP, which enhances the exchange of knowledge with the external world by opening up organizational boundaries. The difference lies in the proactive and strategic learning of the DP, compared with the passive and responsive organizational learning of EPs. EPs, as upstream and downstream companies in the value chain, strategically comply with the DP and choose to participate in specific open-source projects. Organizational learning enables the EPs to enhance their technological and innovative capabilities, as well as to promote VCC with other participants.

**5.2.3 The IPs' OSVCC process.** The IPs are normally independent developers, including enthusiasts and technical experts. The IPs contribute to innovation and diversity, offer their labor and time voluntarily, and foster a culture of learning in the OSCs. Payne, *et al.* [15] suggest that the customer's value co-creating process is characterized by its dynamic, interactive, and nonlinear nature, and often occurs unconsciously. In the OSVCC framework, the IPs' engagement in the relationship experience encompasses emotional, cognitive, and behavioral elements:

1. IPs develop a profound emotional connection to the community, and possess a strong passion for their technical field, willingly contributing their time and energy;

2. IPs effectively leverage their cognitive abilities, encompassing technical and business skills, to enhance the vitality and diversity of OSCs;

3. IPs exhibit self-motivated altruistic behavior, propelling the development and innovation of OSCs through collaborative and active contributions.

The IPs are highly proactive in their pursuit of learning and personal development, and not only improve their own skills but also enhance their professional prospects. The IPs engage in continuous learning while solving problems and completing tasks, actively seeking new knowledge and skills to enhance their contributions and value. The IPs actively participate in various activities and discussions, including conferences, seminars, and workshops, to foster face-to-face communication with other participants. The active involvement of IPs contributes vibrancy and creativity to OSCs.

**5.2.4 Encounter processes.**   Encounter processes are collaborative practices involving the exchange of resources and engagement in activities between suppliers and customers, including communication, use, and service [15]. Encounter processes become more complex in the OSVCC framework because it involves three participants. This study's primary objective was to propose the framework for OSVCC, and thus it does not delve into an in-depth discussion of the encounter processes between the participants, although it is important to acknowledge their presence. The authors argue that the encounter processes between the participants should align with the principles of OSVCC, which include openness, sharing, collaboration, and freedom [7]. Participants should design and implement encounter processes according to these principles. Furthermore, there is a game of interest among the participants during the encounter processes [41].

# 6 Conclusions and implications

The open-source concept has emerged as an important software development model in the IT industry, and the VCC has been widely embraced as a marketing strategy. However, there is a scarcity of research regarding VCC by OSCs. From a theoretical perspective, this research fills the gap in the existing literature. On the one hand, we systematically constructed the research framework of OSVCC and found the triangular relationship. This framework inherits the widely adopted VCC process model and emphasizes a view of the collaboration of the participants. On the other hand, we creatively combined the business ecosystem theory and the open innovation theory to build a four-quadrant classification model of OSCs. This classification model enables us to examine the existing OSCs from two dimensions: maturity and commercialization.

From a practical perspective, this research provides four managerial implications for the enterprise. Firstly, the complexity of the OSVCC process challenges traditional VCC management practices. The relationship between the participants in VCC is no longer the supplier-customer relationship, but the triangular relationship. The enterprise's decision may affect the other two participants and ultimately affect the stability of the triangular relationship. Secondly, the framework shows that the enterprises choose different focuses in OSVCC. If an enterprise is the founder of the OSC, the enterprise should focus on capturing opportunities, formulating plans, implementing, and evaluating. If an enterprise is an OSC user or developer, it should focus on following the domination enterprise, looking for opportunities and taking actions based on its own cognition. Thirdly, this research stresses the importance of organizational learning. No matter what kind of role, an enterprise needs to break through organizational boundaries to learn and exchange knowledge when participating in OSVCC. Finally, the implementation issues of the OSVCC should be of concern. The participants need to understand the process of OSVCC and take the initiative to design and deliver relevant relational experiences. Our research also stresses the importance of organizational learning and knowledge management by managing and designing the encounter process from a long-term perspective.

This study has several limitations that warrant further research and exploration. Firstly, this study employs a static approach, which could not analyze the dynamic development of the cases. It is important to note that a successful OSC typically undergoes a multi-year development process and encounters fluctuations due to internal and external changes. Hence, future research could explore the potential of integrating the framework proposed in this paper to examine the interrelationships among participants, considering the dynamic development perspective. Secondly, this study only examined four Chinese OSCs, but almost all well-known open-source projects are born in the United States. So, the research conclusions may not be

globally representative. The authors would welcome the application of the OSVCC framework to OSC studies in other countries. Thirdly, limited by various factors, this paper does not discuss the three encounter processes in the OSVCC in detail. Future research could address a more thorough analysis of the intricate encounter processes among participants by game theory.

## Acknowledgments

We wish to thank Ericsson China Academy for supporting, and appreciate three anonymous reviewers for their insights and helpful comments on previous versions of this article.

## Author Contributions

**Conceptualization:** Yang Luo, Yongsheng Jin.

**Formal analysis:** Yang Luo, Yuanmao Ji.

**Methodology:** Yang Luo, Yongsheng Jin, Yuanmao Ji.

**Resources:** Yongsheng Jin, Yuanmao Ji.

**Supervision:** Yang Luo.

**Validation:** Yang Luo, Yuanmao Ji.

**Writing – original draft:** Yang Luo.

**Writing – review & editing:** Yongsheng Jin.

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
