## [Decision Letter · Decision Letter 0]

1 Mar 2024

PONE-D-23-37978Explore an Open-Source Value Co-Creation Framework: A Multiple Case Study in ChinaPLOS ONE

Dear Dr. Luo,

Thank you for submitting your manuscript to PLOS ONE. After careful consideration, we feel that it has merit but does not fully meet PLOS ONE’s publication criteria as it currently stands. Therefore, we invite you to submit a revised version of the manuscript that addresses the points raised during the review process.

We look forward to receiving your revised manuscript.

Kind regards,

Silvia Escribano Cubas

Academic Editor

PLOS ONE

Journal Requirements:

3. "We note that your Data Availability Statement is currently as follows: ""All relevant data are within the manuscript and its Supporting Information files.""

If there are ethical or legal restrictions on sharing a de-identified data set, please explain them in detail (e.g., data contain potentially sensitive information, data are owned by a third-party organization, etc.) and who has imposed them (e.g., an ethics committee). Please also provide contact information for a data access committee, ethics committee, or other institutional body to which data requests may be sent. If data are owned by a third party, please indicate how others may request data access

Reviewers' comments:

Reviewer's Responses to Questions

**Comments to the Author**

1. Is the manuscript technically sound, and do the data support the conclusions?

Reviewer #1: Yes

Reviewer #2: Partly

Reviewer #3: Yes

2. Has the statistical analysis been performed appropriately and rigorously? 

Reviewer #1: Yes

Reviewer #2: N/A

Reviewer #3: Yes

3. Have the authors made all data underlying the findings in their manuscript fully available?

Reviewer #1: Yes

Reviewer #2: Yes

Reviewer #3: Yes

4. Is the manuscript presented in an intelligible fashion and written in standard English?

Reviewer #1: Yes

Reviewer #2: Yes

Reviewer #3: Yes

5. Review Comments to the Author

Reviewer #1: Some suggestions to the authors:

1. The authors need to highlight in brief why open-source value cocreation(OSVCC) is very important topic in the abstract

2. It is not necessary to include the sub-section after the conclusion such as 5.1, 5.2 & 5.3.

3. It is good for the authors to summarize the Conclusion and not too details.

Reviewer #2: The paper describes an Open-Source Value Co-Creation Framework based on case studies. Such an open-source value framework may be invaluable to organizations exploring the pros and cons of adopting open-source software in enterprise systems and the engagement with OS teams.

Major comments:

Though the case study is confined to a particular geographic location, an attempt should be made to make the findings generalizable. Such an attempt is conspicuously missing in this approach. Please comment on why the authors consider that the framework applies only to one region. More specifically, how is the Chinese context different from the global context?

This reviewer considers the lack of interviews as a data collection method, as a major limitation.

Theoretical conceptualization to a theory from the described relationship is missing. Perhaps that is the reason for doubts about generalizability.

Minor comments

Prior work in the introduction section needs to be organized better, describing how prior work in this domain led the authors to this research question.

It is not clear how the two-stage model of OSC relates to business ecosystem theory.

The "commercial dimension" in HCM seems to be ill-defined.

Reviewer #3: It’s a pleasure to review this manuscript addressing a significant issue. Some suggestions are proposed as follows.

1. Since value co-creation is the essential concept of this study, it is expected to have more support from the literature.

2. The authors proposed an excellent classification model based on the business ecosystem and open innovation theory; readers would expect some discussion on the rationale and background.

3. The quality aspect of the case study methodology could be addressed.

4. Readers would like more explanations on the proposed conceptual framework of OSVCC.

5. The managerial implications could discuss the implementation issues regarding the value co-creation of OSCs.

6. Authors comment on one of the limitations of their study, which only examined four Chinese OSCs, which might differ from other well-known open source projects born in the United States. After this study, it seems they should have some standpoints.

6. PLOS authors have the option to publish the peer review history of their article (what does this mean?). If published, this will include your full peer review and any attached files.

Reviewer #1: No

Reviewer #2: No

Reviewer #3: No

---

## [Author Response · Author response to Decision Letter 0]

16 May 2024

Dear Editor and Reviewers,

We sincerely appreciate your valuable time spent providing constructive remarks and useful suggestions, which have significantly enhanced the quality of the manuscript and enabled us to make improvements. Each suggested revision and comment, brought forward by the reviewers was accurately incorporated and considered. We have responsed the comments of the reviewers point by point and the revisions are indicated. Please refer to attached file "Response to Reviewers.docx".

---

## [Decision Letter · Decision Letter 1]

7 Jul 2024

PONE-D-23-37978R1Explore an Open-Source Value Co-Creation Framework: A Multiple Case StudyPLOS ONE

Dear Dr. Luo,

Thank you for submitting your manuscript to PLOS ONE. After careful consideration, we feel that it has merit but does not fully meet PLOS ONE’s publication criteria as it currently stands. Therefore, we invite you to submit a revised version of the manuscript that addresses the points raised during the review process.

We look forward to receiving your revised manuscript.

Kind regards,

Silvia Escribano Cubas

Academic Editor

PLOS ONE

Journal Requirements:

Reviewers' comments:

Reviewer's Responses to Questions

**Comments to the Author**

1. If the authors have adequately addressed your comments raised in a previous round of review and you feel that this manuscript is now acceptable for publication, you may indicate that here to bypass the “Comments to the Author” section, enter your conflict of interest statement in the “Confidential to Editor” section, and submit your "Accept" recommendation.

Reviewer #1: All comments have been addressed

Reviewer #3: (No Response)

2. Is the manuscript technically sound, and do the data support the conclusions?

Reviewer #1: Yes

Reviewer #3: Partly

3. Has the statistical analysis been performed appropriately and rigorously? 

Reviewer #1: Yes

Reviewer #3: Yes

4. Have the authors made all data underlying the findings in their manuscript fully available?

Reviewer #1: Yes

Reviewer #3: No

5. Is the manuscript presented in an intelligible fashion and written in standard English?

Reviewer #1: Yes

Reviewer #3: No

6. Review Comments to the Author

Reviewer #1: To improve some of the citations especially for the article having more than 2 authors. For example Payne, Storbacka and Frow (2008) should be replaced with Payne et al. (2008)

Reviewer #3: It’s a great pleasure to review this manuscript, which addresses issues I am interested in. My primary concerns are as follows.

1. As value co-creation (VCC) is the core concept of this study, a solid theoretical background could be supported first. Then come the features and challenges of open-source communities that lead to the introduction of the OSVCC

2. Since VCC is an essential part of service-dominant logic, the authors could comment on the differences between service-dominant, customer-dominant, and OSVCC.

3. VCC models could be presented in a more organized manner, stressing this study's differentiated advantages.

4. The authors could defend their combining use of the business ecosystem theory and the open innovation theory with in-depth elaborations.

5. The case selection criteria might be vague and ambiguous; therefore, some concrete requirements are suggested.

6. The comparative analyses could be conducted more formally and systematically.

7. The conclusions could stress the theoretical and managerial implications more significantly.

7. PLOS authors have the option to publish the peer review history of their article (what does this mean?). If published, this will include your full peer review and any attached files.

Reviewer #1: No

Reviewer #3: No

---

## [Author Response · Author response to Decision Letter 1]

22 Aug 2024

Dear reviewers,

On behalf of all authors, I sincerely appreciate your valuable time spent providing constructive remarks and useful suggestions, which have significantly enhanced the quality of the manuscript and enabled us to make improvements. Each suggested revision and comment, brought forward by the reviewers was accurately incorporated and considered. We have responded the comments of the reviewers point by point and the revisions are indicated. Please refer to attached file "Response to Reviewers.docx".

Sincerely

Yang Luo

---

## [Editor Report · Decision Letter 2]

3 Sep 2024

Explore an Open-Source Value Co-Creation Framework: A Multiple Case Study

PONE-D-23-37978R2

Dear Luo,

We’re pleased to inform you that your manuscript has been judged scientifically suitable for publication and will be formally accepted for publication once it meets all outstanding technical requirements.

Kind regards,

Silvia Escribano Cubas

Academic Editor

PLOS ONE

---

## [Editor Report · Acceptance letter]

11 Sep 2024

PONE-D-23-37978R2 

PLOS ONE

Dear Dr. Luo, 

I'm pleased to inform you that your manuscript has been deemed suitable for publication in PLOS ONE. Congratulations! Your manuscript is now being handed over to our production team.

Kind regards, 

on behalf of

Dr. Silvia Escribano Cubas 

Academic Editor

PLOS ONE